# Functional Abdominal Bloating and Gut Microbiota: An Update

**DOI:** 10.3390/microorganisms12081669

**Published:** 2024-08-14

**Authors:** Salvatore Crucillà, Federico Caldart, Marco Michelon, Giovanni Marasco, Andrea Costantino

**Affiliations:** 1Gastroenterology B Unit, Pancreas Center, University of Verona, 37134 Verona, Italy; federicocaldart94@gmail.com; 2Department of Pathophysiology and Transplantation, University of Milan, 20122 Milan, Italy; marco.michelon@unimi.it; 3Department of Medical and Surgical Sciences, University of Bologna, 40138 Bologna, Italy; giovannimarasco89@gmail.com; 4IRCCS Azienda Ospedaliero-Universitaria di Bologna, 40138 Bologna, Italy; 5Unit of Gastroenterology and Endoscopy, Foundation IRCCS Ca’ Granda Ospedale Maggiore Policlinico, 20122 Milan, Italy

**Keywords:** functional abdominal bloating, gut microbiota, microbioma, functional distension

## Abstract

(1) Background: Functional abdominal bloating and distension (FAB/FAD) are common disorders of the gut–brain interaction. Their physiopathology is complex and not completely clarified, although gut microbiota imbalances play a central role. The treatment of FAB/FAD still represents a clinical challenge for both patients and healthcare providers. Gut microbiota modulation strategies might play a crucial role in their management. The aim of this narrative review was to update the current evidence on FAB/FAD, with a focus on gut microbiota. (2) Methods: In October 2023, a review was conducted through the Medline, PubMed, and Embase databases. Selected literature included all available English-edited studies (randomized controlled trials and cross-sectional, cohort, and case-control studies). (3) Results: Twelve studies were selected, most of which investigated the relationship between IBS and microbiota, with bloating being one of its symptoms. The studies suggest that restoring a balanced microbiome appears to be the most promising solution for better management of FAB/FAD. Targeted approaches, such as the use of probiotics, prebiotics, antibiotics such as rifaximin or dietary modifications, may hold the key to alleviating symptoms. Other therapeutic options, such as diet, neuromodulators, and brain–gut behavioral therapies (i.e., cognitive-behavioral therapy) have shown promising outcomes, but strong data are still lacking. (4) Conclusions: Targeted approaches that focus on the gut microbiota, such as the use of probiotics, prebiotics, and antibiotics, are essential in managing FAB/FAD. Understanding the complex relationship between gut microbiota and FAB/FAD is crucial for developing effective treatments. Further studies are needed to explore the specific roles of different microbial populations in patients with FAB/FAD to enhance therapeutic strategies.

## 1. Introduction

### 1.1. Overview of Abdominal Bloating and Distension

Abdominal bloating and distension are common symptoms associated with disorders of the gut–brain interaction (DGBIs). Bloating is defined by subjective sensations of trapped gas, abdominal pressure, and premature fullness, whereas distension is manifested as quantifiable increases in abdominal girth [1]. In accordance with the Rome IV criteria, functional abdominal bloating (FAB) and distension (FAD) may occur concurrently or independently and are frequently concomitant with functional dyspepsia, irritable bowel syndrome, and functional constipation [2,3]. The FAB/FAD entity is regarded as a single entity despite comprising two distinct symptoms: recurrent abdominal fullness, pressure, or gas sensation (FAB) and/or objective, measurable abdominal girth increase (FAD) occurring at least once per week for a minimum of eight weeks [4]. Patients must not meet the criteria for a diagnosis of another DGBI. However, mild abdominal pain and/or minor abnormalities in bowel movements may coexist. Symptoms should precede diagnosis by a minimum of six months [1,5]. Bloating and distension affect approximately 30% of the general adult population [6], with a prevalence of FAB/FAD of approximately 3.5% and 1.2%, respectively [7]. Patients report significantly impaired quality of life, with 75% experiencing moderate to severe symptoms and 50% facing limitations in daily activities.

### 1.2. Pathophysiology and Contributing Factors

The pathophysiology of FAB/FAD is intricate and not yet fully elucidated, with a multitude of contributing factors. Several factors have been put forth as potential contributors to the development of abdominal bloating and distension, including alterations in the gut–brain axis resulting from changes in the intestinal microbiome. The key contributors to FAB/FAD encompass abnormal intraluminal content, impaired abdominal emptying, altered intestinal gas handling, and visceral hypersensitivity [8,9,10,11]. Furthermore, abdomino-phrenic dyssynergia and pelvic floor dysfunction may also be involved in the etiology of these symptoms [1,12,13,14,15,16,17,18,19,20,21]. Dietary factors, particularly those of a high-osmolarity and poorly absorbable nature, such as FODMAPs, serve to exacerbate the symptoms. Psycho-emotional factors, including stress and depression, have also been identified as significant contributors to the condition [14,22,23,24,25].

### 1.3. Diagnostic Challenges and Treatment Approaches

The treatment of FAB/FAD presents a significant clinical challenge due to the absence of a standardized, effective regimen. The initial evaluation of a patient with suspected FAB/FAD typically begins with a comprehensive medical history, a physical examination, and the performance of appropriate diagnostic tests to rule out organic causes such as celiac disease, malabsorptive disorders, gut dysmotility, or chronic intestinal pseudo-obstruction [26,27,28]. In the initial stages of treatment, non-pharmacological approaches, such as dietary modifications (e.g., low-FODMAP or gluten-free diets), are often employed [25]. Probiotics and antibiotics have been the subject of investigation, with probiotics potentially modulating the gut microbiome to alleviate bloating, whereas antibiotics such as rifaximin may reduce bacterial gas production [29,30,31,32,33]. Pharmacological options, including antispasmodics, secretagogues, prokinetics, and neuromodulators, are employed in clinical practice, though with varying response rates [1,34]. The objective of this narrative review is to synthesize the current evidence on functional bloating and distension, with a particular emphasis on the potential involvement of the microbiome in their pathogenesis and the implications for therapeutic intervention.

## 2. Methods

### 2.1. Literature Search

We conducted a narrative review. PubMed and Embase were searched until October 2023 for studies reporting on FAB/FAD and microbiota or microbiome. A search strategy with the following keyword combinations was used: “functional bloating”; “functional distension”; “abdominal bloating”; “abdominal distension”; “microbiota” and “microbioma”. No time restrictions were applied. Articles that did not have an English translation available were excluded. Further eligible studies were extracted from a review of reference lists of full texts retrieved after the initial screening of search results.

### 2.2. Study Selection and Data Extraction

Study references and citations were collected in Rayyan platform. After removing duplicates, a double-blind selection was carried out by the two main authors (SC and FC) by reading titles, abstracts, and full texts in case of doubts. Study designs eligible for inclusion were randomized controlled trials, cross-sectional studies, cohort studies, case-control studies and case series. Unpublished abstracts, letters, posters, and grey literature were excluded.

At the end of this screening, double blinding was deactivated. Disagreements or conflicting cases between the two authors were resolved by consulting senior authors (MM, AC). Manuscripts potentially inherent to our research were assessed in full text. Extracted data were transferred to Microsoft Excel Version 16.87 (24071426) (Microsoft, Redmond, DC, USA) for interpretation.

### 2.3. Study Quality Assessment

The quality assessment of studies eligible for inclusion was evaluated using the modified Newcastle–Ottawa Scale (NOS). Studies with NOS scores of 0–3, 4–6, and 7–9 were considered as low (L), average (A), and high (H) quality, respectively. Case reports and series were considered low quality.

### 2.4. Institutional Review Board Approval and Informed Consent

Data extraction and review were carried out using previously published studies. Accordingly, no Institutional Review Board approval or patient informed consent were required.

## 3. Results

### 3.1. The Multifactorial Pathophysiology of Functional Abdominal Bloating and Distension: The Contributing Factors

We found two studies on abdominal bloating/distention and microbiota. Most of the studies were on IBS and microbiota, including bloating as a symptom, and only a few investigated the relationship between microbiota and abdominal bloating [35,36].

The causes of chronic abdominal bloating and distension are complex and often depend on multiple factors. Therefore, the pathophysiology of these conditions is not fully understood, and emerging evidence suggests the role of many elements, especially of microbiota (Figure 1).

FAB/FAD can be caused by various factors, including an abnormal increase in intraluminal content (such as gas, air, water, and fecal material within the gut), impaired abdominal emptying, and impaired intestinal handling of gas loads, even with normal gas volumes [12,13,14,15,37,38]. Additionally, a functional or perceptive alteration of the bowel symptoms can result in a visceral hypersensitive alteration [8,9,10,11]. Furthermore, there may be an altered intra-abdominal volume displacement: the gas load induces an abnormal redistribution of luminal content due to a paradoxical relaxation of the anterior wall and a paradoxical contraction of the diaphragm, leading to an abnormal response to gas load [16].

Other causes might include abdomino-phrenic dyssynergia and pelvic floor dysfunction, caused by a paradoxical response of the diaphragm and anterior abdominal wall muscles [1,17,18,19,20].

Symptoms induced by diet may result from increased sensitivity to bowel distension, which is particularly aggravated by the consumption of high-osmolarity, poorly absorbable foods such as FODMAPs [25]. It has been hypothesized that individuals with the FC-H catabotype may experience more pronounced benefits from adhering to a low-FODMAP diet. The PROBE-IBS/2 trial findings support the idea that adhering to dietary recommendations leads to a significant reduction in abdominal pain and an improvement in stool consistency in this subgroup [24].

Psycho-emotional factors, such as stress and depression, may contribute to increased perception of abdominal bloating [1,2,3,4]. The frequency and severity of bloating, in conjunction with the presence of a DGBI, have an impact on both psychological and physical well-being, as suggested by the results of pairwise comparisons. Individuals experiencing bloating exhibited more severe depression and anxiety, reduced overall mental health, and decreased overall physical health. Additionally, the presence of a DGBI was associated with higher levels of psychological and physical health burden [19,20].

### 3.2. The Role of Microbiota in the Physiopathology of Bloating and Distension

Emerging, but still unreliable, evidence derived from studies on DGBIs has suggested an interesting role of the microbiome and its alteration in the development of bloating as a symptom, not as a nosological entity defined as Rome IV criteria [3]. To date, no studies have specifically examined the relationship between the gut microbiome and the development of FAD/FAB.

However, many studies have investigated the correlation between gut microbiota and motility and intestinal permeability in gastrointestinal disorders [5,21,22,23,24,33,39,40].

Before the introduction of the Rome IV criteria [3], different studies, especially on IBS [41,42,43,44], reported that an increased or decreased production of colonic gas, leading to bloating, is related to an alteration in colonic microbiota through a fermentation process. Furthermore, inflammation caused by an alteration of host-microbiota leads to gut sensory and motor dysfunction which may contribute to bloating [45]. In addition, Vernia et al. [46] described those patients with a low production of methane as having increased bloating following the ingestion of sorbitol and fibers. Hence, differences in the number and types of intestinal microorganisms have been observed, comparing patients with IBS to healthy individuals [47].

Emerging data [37] showed a reduced microbial abundance complexity and diversity [35,41] and an alteration in phyla composition, with an increase in *Bacteroidetes* and *Clostridia* and a reduction in *Bifidobacterium*. A cohort study by Ringel-Kulka [35] described that the *Ruminococcaceae* were abundant taxa in IBS subjects with bloating.

A meta-analysis of 17 RCTs showed a significant effect of probiotics on bloating scores [41]. An international guide through a systematic review and consensus voting reported 70% agreement and a moderate grade of evidence for the effect of certain probiotics on bloating in people with IBS [34,48].

The loss of microbial richness and diversity which has been observed in patients with IBS contributes to the pathogenesis of bloating through alterations in intestinal neuromuscular function, mucosal cell junction integrity, immune profile, and inflammatory response. This suggests a weakness in epithelial barrier functions that could partially explain the cause of bloating in patients with IBS [49]. However, scant and sparse data are available on gut microbiota signatures in patients with FAB/FAD and most of the information is permeated by studies including IBS patients.

### 3.3. Therapeutic Options for Bloating and Distension: Diet

Food and gut microbiota act together influencing many host functions such as nutrition, immunity, and the enteric nervous system [34]. The role of food and diet in producing intestinal gas and triggering abdominal distension has been extensively studied, primarily among patients with IBS, as dietary modifications or eliminations have been suggested as part of the treatment [49]. The low-FODMAP diet appears to be the most effective dietary strategy for improving IBS symptoms, including abdominal pain and bloating. A recent systematic review and meta-analysis demonstrated the superiority of the low-FODMAP diet compared to other interventions in alleviating symptoms in IBS patients, including bloating and abdominal distension [50].

The traditional dietary advice (TDA), developed by the National Institute for Health and Care Excellence (NICE) and the British Dietetic Association (BDA), is considered another first-line dietary therapy for IBS. Its principles include having regular meals, adjusting fiber and fluid intake, and reducing consumption of fat, alcohol, and caffeine [51].

In 2015, Böhn et al. conducted a randomized controlled trial on 75 patients meeting the Rome III criteria for IBS demonstrating that the traditional dietary advice reduced the severity of IBS symptoms as well as a low-FODMAP diet (*p* < 0.0001 in both groups before vs. at the end of a 4-week diet) [52].

However, due to the complexity of the low-FODMAP diet, the risk of associated nutritional deficiencies requiring close monitoring by a gastroenterology dietitian [53,54], and the potential adverse effects on the gut microbiota including a reduction in Bifidobacterium species [55,56], many guidelines suggest using TDA as a primary approach for the management of IBS and the low-FODMAP diet as a second-line dietary therapy [48].

Direct studies on the effect of a low-FODMAP diet in functional bloating and distension have not yet been conducted. However, given the results observed in IBS patients, dietary intervention could potentially play a role in future treatments for FAB/FAD.

Regarding the gluten-free diet (GFD), randomized controlled trials have presented conflicting data about its efficacy in controlling IBS symptoms. Additionally, there is insufficient scientific evidence to recommend it for the management of bloating and abdominal distension [57].

### 3.4. Therapeutic Options for Bloating and Distension: Modulation of the Microbiota

Despite FAB/FAD, many other gastrointestinal diseases present with bloating and abdominal distension which represent two of the most reported symptoms during gastroenterological examination [58]. Due to limited scientific literature on the subject, currently, no validated treatment algorithm exists.

One way to modulate the gut microbiota involves enhancing the population of beneficial bacteria using prebiotics and probiotics [59]. Probiotics are among the most prescribed products for bloating even though literature reports conflicting data about their real efficacy.

The general benefit of probiotics on the gut microbiota consists of creating a more favorable gut environment through mechanisms shared by most probiotics. The two most common general benefits induced by probiotics are supporting a healthy digestive tract and a healthy immune system. The widespread mechanisms associated with these benefits include the inhibition and competitive exclusion of potential pathogens, regulation of intestinal transit, normalization of perturbed microbiota, and production of useful metabolites (e.g., acids and short-chain fatty acids) or enzymes [60].

In 2011, Ringel-Kulka et al. conducted a double-blind, placebo-controlled clinical trial among sixty patients meeting the Rome III criteria for non-constipation IBS, functional diarrhea, or functional bloating. The group that took *Lactobacillus acidophilus NCFM* and *Bifidobacterium lactis Bi-07* twice daily for 8 weeks showed an improvement in abdominal bloating severity compared to the placebo group, both at 4 and 8 weeks (*p* = 0.02, *p* < 0.01 respectively) [30].

A multicenter, double-blind, randomized, placebo-controlled, parallel study with a 2-week placebo run-in phase followed by a 4-week intervention phase, enrolling 302 patients experiencing abdominal discomfort and bloating ≥2 times per week for at least three months without a medical evaluation or drug prescription in the previous 12 months, demonstrated that the administration of *Bifidobacterium infantis* 35624 [61] did not show differences in bloating relief among the enrolled subjects.

Another randomized control trial (RCT) of a multi-strain probiotic mix of *Streptococcus thermophilus* BT01, *Bifidobacterium breve* BB02, *Bifidobacterium longum** BL03, *Bifidobacterium infantis** BI04, *Lactobacillus acidophilus* BA05, *Lactobacillus plantarum* BP06, *Lactobacillus paracasei* BP07, and *Lactobacillus delbrueckii* subsp. *Bulgaricus*** BD08 vs. placebo showed a borderline significant difference in abdominal bloating scores (*p* = 0.09), with a reduction in abdominal bloating in the multi-strain group (*p* = 0.046) [mean post-minus pre-treatment score, −13.7; 95% confidence interval (CI), −2.5 to −24.9] [44]. In contrast, another study conducted using the same multi-strain probiotics demonstrated a reduction in flatulence but no effect on bloating [62].

A recent multicenter, randomized, double-blind, placebo-controlled study conducted on sixty-six patients meeting the Rome IV diagnostic criteria for functional bloating or distension demonstrated a significant reduction in bloating symptoms in the group taking *Bacillus coagulans* MTCC 5856 daily for 4 weeks [63].

Most studies related to bloating treatment have been conducted on IBS patients, so currently, there is insufficient scientific evidence to recommend the use of probiotics in patients exclusively affected by functional bloating or distension. Therefore, the most recent European and American guidelines do not recommend the use of probiotics for bloating treatment [64].

The use of antibiotics to modulate the gut microbiota in DGBI has been widely studied in patients affected by IBS or small intestine bacterial overgrowth (SIBO) [65,66]. The rationale for their use consists of reducing the total number of intestinal bacteria to decrease the production of luminal gas.

The most studied is rifaximin, a non-absorbable gastrointestinal-specific oral antibiotic, with a broad spectrum of activity and a good safety profile [67]. This antibiotic is also able to exert a direct anti-inflammatory effect modulating pregnane X receptor (PXR), therefore acting directly on low-grade inflammation [68]. Rifaximin has been extensively studied in IBS patients, demonstrating a significant improvement in symptoms, including abdominal bloating [69]. Its effectiveness in this context is believed to stem from the modification of the gut microbiota, resulting in reduced mucosal inflammation and enhanced barrier function in the small bowel [70]. A recent systematic review conducted by Arora et al. demonstrated that rifaximin therapy administered in doses from 400 to 1650 mg per day for 1–2 weeks was associated with an increased likelihood of improvement in abdominal bloating and distension, reducing the severity of these symptoms in patients with DGBI, even though daily doses less than 1200 mg/day were similar to a placebo [71].

Two identically designed, phase 3, double-blind, placebo-controlled trials (TARGET 1 and TARGET 2), which enrolled 1260 patients affected with IBS without constipation, showed a significant relief of bloating (39.5% vs. 28.7%, *p* = 0.005, in TARGET 1; 41.0% vs. 31.9%, *p* = 0.02, in TARGET 2; 40.2% vs. 30.3%, *p* < 0.001, in the two studies combined) in patients randomly taking rifaximin at a dose of 550 mg three times daily for 2 weeks, along with concurrent relief from IBS symptoms, abdominal pain, and loose or watery stools [72].

### 3.5. Therapeutic Options for Bloating and Distension: Neuromodulators

DGBI are often associated with visceral hypersensitivity which can be considered as a potential cause of bloating conditions [73].

The use of neuromodulators such as antidepressants in the treatment of functional gastrointestinal symptoms and disorders is widespread [74]. However, no clinical trial has specifically focused on bloating as a primary symptom. Therefore, the efficacy of these treatments on abdominal bloating has consistently been evaluated as a secondary endpoint in trials conducted on patients affected by IBS and other functional gut disorders like FD [75].

In a small double-blind, randomized clinical trial conducted on 33 patients with moderate to severe IBS, the group receiving venlafaxine, a serotonin-norepinephrine reuptake inhibitor, showed a significant improvement in the severity of all gastrointestinal symptoms (including bloating and distensions), depression, anxiety, stress, and quality of life (QoL) [76].

Another multicenter, prospective trial randomized FD patients to 12 weeks of amitriptyline (50 mg), escitalopram (10 mg), or a matching placebo. In this study, patients treated with both escitalopram and amitriptyline showed an improvement in postprandial bloating [75].

On the other hand, in a recent randomized, double-blind, placebo-controlled trial involving 463 patients with various types of IBS and ongoing symptoms, Ford et al. demonstrated that titrated low-dose amitriptyline was superior to a placebo as a second-line treatment for IBS. However, although significantly more participants allocated to amitriptyline experienced a 30% or greater decrease in abdominal pain severity on the IBS Severity Scoring System (IBS-SSS) at 6 months (1.66, 1.12–2.46; *p* = 0.012), there was no significant difference observed in abdominal distension severity [77].

According to American Gastroenterology Association (AGA) guidelines, antidepressants and other central neuromodulators can be employed for the treatment of bloating and abdominal distension [64]; these medications work by reducing visceral hypersensitivity, increasing sensation threshold, and addressing psychological comorbidities [74].

Considering other therapies, which is beyond the aim of our review, smooth muscle antispasmodics may improve symptoms if they arise due to gaseous distension of the intestinal tract, secretagogues as lubiprostone and linaclotide are also able to improve bloating. Data on the use of prokinetics for the treatment of chronic bloating and distension are limited. Prucalopride improved symptoms of bloating [1].

### 3.6. Therapeutic Options for Bloating and Distension: Other Interventions

Brain–gut behavioral therapies, such as cognitive-behavioral therapy (CBT) and hypnotherapy, are psychological therapies that have shown efficacy in treating IBS symptoms such as anxiety and depression [78]. Although no therapy has specifically targeted functional bloating alone, experts suggest their use, at least in combination with other treatments, to alleviate the burden associated with the condition [64]. Other studied strategies for the treatment of functional bloating include anorectal biofeedback when a pelvic floor disorder such as dyssynergia in defecation is identified [79].

## 4. Discussion

We aimed to review the recent evidence on the physiopathology and treatment of FAB/FAD, with a focus on the role of the microbiome and its potential modulation.

Emerging data show a complex interaction between the main intestinal factors, including the intestinal barrier, bowel motility, microbiome composition, and neurological factors such as brain processing and visceral hypersensitivity. Many therapeutic approaches have been investigated but, to date, there is not a standardized consistent therapeutic algorithm for FAB/FAD.

Since the introduction of the Rome IV criteria in 2016, few studies have investigated the true prevalence and therapeutic options for FAB/FAD as distinct entities from the other DGBIs. Usually, bloating and distension coexist with other gastroenterological symptoms, such as constipation, diarrhea, abdominal pain and are consequently included in DGBIs. Therefore, patients with isolated FAD/FAB, despite its prevalence in approximately 3% of the general population [26], are not adequately studied. 

The microbiome may play a central role in sustaining gastrointestinal health with its intricate and diverse composition of microorganisms. Recent studies delving into the microbiomes of individuals grappling with abdominal bloating and distension have unveiled noteworthy alterations in microbial composition and alpha/beta diversity; these findings indicate a potential involvement in the pathogenesis of these symptoms [33,50,51]. Notably, a decreased microbial diversity is often observed, along with an imbalance in the abundance of specific bacterial taxa. This includes an increase in *Bacteroides* and *Clostridia*, whereas *Bifidobacterium* and *Lactobacillus* are found in reduced quantities in individuals with these symptoms [37,38,41,80,81]. These changes can potentially disrupt intestinal neuromuscular function and compromise mucosal cell junction integrity. This may lead to altered immune responses and the initiation of inflammatory processes, all of which may contribute to the exacerbation of abdominal bloating and distension.

Some authors have proposed that H2 plays a regulatory role in fermentation processes within the human gut microbiome. The results demonstrate that H2 production and consumption exert a significant influence on the production of butyrate, an anti-inflammatory metabolite. This underscores the potential for microbial therapies targeting H2 levels and butyrate production to manage functional abdominal bloating and distension [82]. Further research is required to develop more effective treatments for these conditions based on these strategies.

Despite the lack of a high level of evidence, the role of the microbiome in the management of FAB/FAD is likely crucial.

Diet, probiotics, prebiotics, and antibiotics have not been investigated enough as therapeutic options for modulating the gut microbiota and alleviating symptoms of FAB/FAD. Although probiotics have shown promise in some studies in patients with DGBIs, their efficacy for FAB/FAD remains uncertain due to conflicting data. Moreover, a possible bias can be represented by the different types and dosage used in the different studies. In the near future, modifications to the gut microbiota with specific bacterial strains (single or multi-strain) or other microbiota-targeted options available (e.g., non-absorbable antibiotics or fecal microbiota transplantation) deserve further prospective studies for patients with FAB/FAD, particularly for long-term effects.

The advent of novel, next-generation, live biotherapeutics such as *Faecalibacterium Prausnitzii* [83] and *Akkermansia Muciniphila* [84], which have also shown positive effects in patients with DGBI, is also going to be disruptive for the microbiota modulation in these patients.

Future research should concentrate on the development of bespoke therapies that are specifically designed for the gut microbiome. The current limitations in this field include a lack of comprehensive understanding of the complex interactions between various microbial species and their metabolic pathways. Additionally, the variability in microbiome composition between individuals presents a challenge to the standardization of treatments. Advancing this field requires further studies to identify precise microbial targets and effective interventions, as well as longitudinal research to assess the long-term efficacy and safety of these tailored therapies.

## 5. Conclusions

Abdominal bloating and distension are complex and challenging gastrointestinal symptoms which may have a deep impact on the quality of life of many patients, and the lack of high level of evidence-based indications is challenging for their management. Therefore, continued research efforts are necessary to advance our understanding of these conditions. Collaborative and evidence-based approaches are crucial in addressing the needs of patients experiencing abdominal bloating and distension. Their underlying causes are multifaceted and, to date, remain not fully explained.

The intricate interplay between abdominal bloating, distension, and the gut microbiota is an area of growing interest in the medical field; further evidence has been published to suggest a compelling connection between these distressing symptoms and gut microbiota. The emerging insights into these microbiota-related factors strongly emphasize the necessity for further investigations. Understanding the precise mechanisms behind this intricate relationship is pivotal, especially when considering potential therapeutic interventions. Today, the road to restoring a balanced microbiome appears the most promising solution for the better management of FAB/FAD and targeted approaches, such as the use of probiotics, prebiotics, or dietary modifications, may hold the key to alleviating symptoms. Although much remains to be uncovered, the link between these symptoms and the gut microbiota composition and diversity opens doors for future research and the development of targeted therapies, ultimately providing relief for individuals suffering from these distressing symptoms.

To summarize, patients with FAB/FAD can be managed with a comprehensive approach that includes the use of microbiota-targeted treatments, neuromodulators, specific diets, and, when necessary, psychotherapy. It is recommended that a tailored approach be taken with regard to each patient, with particular attention paid to the efficacy, safety, and long-term outcomes of the management of bloating and distension.

## Figures and Tables

**Figure 1 microorganisms-12-01669-f001:**
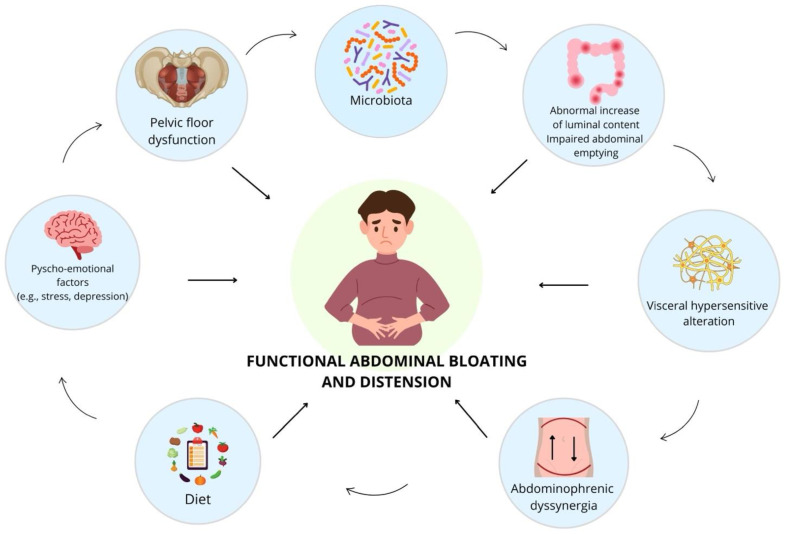
Functional abdominal bloating and distension: a multifactorial pathophysiology.

## Data Availability

The original contributions presented in the study are included in the article, further inquiries can be directed to the corresponding author.

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
