# Peer review of "Functional Abdominal Bloating and Gut Microbiota: An Update"

_microorganisms, 2024, doi:10.3390/microorganisms12081669_

Round 1

Reviewer 1 Report

Comments and Suggestions for Authors

Please consider the following suggestions for revision:

1. The topic of the paper is "Relationship between Functional Abdominal Bloating and Gut Microbiota". The conclusion in the abstract that "Targeted approaches, such as the utilization of probiotics, antibiotics, and neuromodulators are crucial in the management of FAB/FAD. Further studies on gut microbiota are needed for patients with FAB/FAD" is not closely aligned with the main theme. The conclusion should focus more directly on the relationship between functional abdominal bloating and gut microbiota.

2. The Introduction section contains numerous short paragraphs. It is recommended to reorganize the thoughts and logic into three main sections.

3. The methods section is described too briefly. It should be detailed further.

4. The section "2.2. Study selection and data extraction" should include a flowchart for clearer presentation to the readers.

5. The subheading "3.1" in the Results section is too brief. Please refine it into a longer, more descriptive subheading that reflects its content.

6. The figure's legend is absent.

7. A figure illustrating the relationship between Functional Abdominal Bloating and gut microbiota should be included.

Comments on the Quality of English Language

Minor editing of English language required

Reviewer 2 Report

Comments and Suggestions for Authors

The manuscript by Salvatore Crucillà et al. summarized recent advances in the study of functional abdominal bloating and gut microbiota. The study is informative and interesting. I have the following questions and comments:

1, a recent paper (PMID: 37322527) by Austin Campbell et al. studied the production of H2 by the human gut microbiota. H2 production is closely related to functional abdominal bloating and this paper should be discussed. 

2, the authors should add a new table to summarize the effect of different human gut microbes in the production of gases in the colon. Which bacteria produces H2? Which bacteria produces CH4 etc. 

3, future research directions and current limitations in this field should be discussed. 

Reviewer 3 Report

Comments and Suggestions for Authors

This was an interesting study that aims to increase our understanding of gut bloating and distension, a problem that affects many patients..  The paper is well written, but there were some uncertainties about the limited number of studies included.  The following should be addressed:

The aim of our narrative review was to update the evidence on functional bloating and distension, focusing on the possible role of microbiome in their development and its importance for their treatment.

I read the article twice in places, and am still not sure what is the best option to resolve bloating and distension.  A summary at the end of the article of its overall findings would help to clarify this uncertainty.

2.3. Study quality assessment:   The quality assessment…

More explanation of why the inclusion criteria were chosen as stated, as only 12 studies were eventually included?   It would help in the Bibliography of 78 references to know which studies were chosen.

“Psycho-emotional factors, such as stress and depression, may contribute to increased perception of abdominal bloating.”

This is an important point, so this statement needs a reference or 2.

The legend to 3.2 needs expanding to give a clearer overview of the overall conclusion of the multi-panel figure.

Page 5

Emerging, but still unreliable, evidence has suggested an interesting role of the microbiome and its alteration in the development of abdominal bloating and distension.

Followed by:

To date, no studies have specifically examined the relationship between the gut microbiome and the development of abdominal bloating or distension.

How can there be emerging evidence if no specific studies?  Confusing – and so needs clarifying.

Round 2

Reviewer 2 Report

Comments and Suggestions for Authors

The authors have revised the manuscript accordingly. It can be considered for publication. 

Reviewer 3 Report

Comments and Suggestions for Authors

Thanks to the authors for their revisions of the manuscript, and their replies to my comments.  The revised paper now reads much more clearly.